# No Women’s Land: Australian Women Veterans’ Experiences of the Culture of Military Service and Transition

**DOI:** 10.3390/ijerph21040479

**Published:** 2024-04-15

**Authors:** Sharon Lawn, Elaine Waddell, Louise Roberts, Pilar Rioseco, Tiffany Beks, Tiffany Sharp, Liz McNeill, David Everitt, Lee Bowes, Dylan Mordaunt, Amanda Tarrant, Miranda Van Hooff, Jonathan Lane, Ben Wadham

**Affiliations:** 1Flinders University, Adelaide, SA 5042, Australia; elaine.waddell@flinders.edu.au (E.W.); louise.roberts@flinders.edu.au (L.R.); liz.mcneill@flinders.edu.au (L.M.); dylan.mordaunt@flinders.edu.au (D.M.); ben.wadham@flinders.edu.au (B.W.); 2Open Door Initiative, Flinders University, Adelaide, SA 5042, Australia; tiffany.beks@ucalgary.ca (T.B.); tiffany.sharp@cambrianexecutive.com (T.S.); djeveritt@bigpond.com (D.E.); jameslee.bowes@bigpond.com (L.B.); amanda.tarrant2@sa.gov.au (A.T.); 3Lived Experience Australia, Adelaide, SA 5042, Australia; 4Australian Institute of Family Studies, Melbourne, VIC 3006, Australia; pilar.rioseco@aifs.gov.au; 5School of Education, University of Calgary, Calgary, AB T2N 1N4, Canada; 6Cambrian Executive, Adelaide, SA 5000, Australia; 7Defence Force Welfare Association SA, Adelaide, SA 5000, Australia; 8Southern Adelaide Local Health Network, Adelaide, SA 5042, Australia; 9Veterans SA, Adelaide, SA 5000, Australia; 10Military and Services Health Australia (MESHA), Adelaide, SA 5011, Australia; miranda.vanhooff77@gmail.com; 11Department of Psychiatry, University of Tasmania, Hobart, TAS 7005, Australia; jonathan.lane@utas.edu.au

**Keywords:** women, veterans, gender, transition, identity, culture, systems, military sexual trauma, mental health

## Abstract

Women’s experiences of military service and transition occur within a highly dominant masculinized culture. The vast majority of research on military veterans reflects men’s experiences and needs. Women veterans’ experiences, and therefore their transition support needs, are largely invisible. This study sought to understand the role and impact of gender in the context of the dominant masculinized culture on women veterans’ experiences of military service and transition to civilian life. In-depth qualitative interviews with 22 Australian women veterans elicited four themes: (1) Fitting in a managing identity with the military; (2) Gender-based challenges in conforming to a masculinized culture—proving worthiness, assimilation, and survival strategies within that culture; (3) Women are valued less than men—consequences for women veterans, including misogyny, sexual harassment and assault, and system failures to recognize women’s specific health needs and role as mothers; and (4) Separation and transition: being invisible as a woman veteran in the civilian world. Gendered military experiences can have long-term negative impacts on women veterans’ mental and physical health, relationships, and identity due to a pervasive masculinized culture in which they remain largely invisible. This can create significant gender-based barriers to services and support for women veterans during their service, and it can also impede their transition support needs.

## 1. Introduction

Women veterans learn to become military personnel in a male-dominated environment and culture in which their presence is highly visible, challenged, and often subject to institutional prejudice [1,2]. Women veterans experience military sexual trauma (MST) commonly during their service, which, combined with other challenges in the role, can have significant adverse impacts on their transition from the military to civilian life. A Canadian study [3] and an Australian study [2] found female service members experience significant role confusion and often do not report MST for fear of further victimization, blame, and trauma within a dominant masculine military culture. They constantly face low-level sexual innuendo and misconduct and must defer to, resist, or embody military masculine culture to sustain themselves, but this may lead them to avoid seeking help or disclosing MST during service and post-release from the military [4]. A longitudinal US study of 554 female Iraq and Afghanistan War veterans revealed that deployment sexual harassment was the strongest predictor of decreased psychosocial functioning across all domains [5].

Military–civilian transition is internationally recognized as a priority for public policy and research [6,7]. Transition can be confusing and isolating for women veterans, particularly because they must navigate a new identity and roles [3,8]; they neither fit in the military due to gendered relations centered on military forms of masculinity, nor civilian life where they are largely misunderstood as “Veterans”. This “no women’s land” is poorly understood [1,2,9]. Support services in the community are “gender blind” (inherently gender biased), with limited understanding of women veterans’ experiences and needs [2], creating further gender inequality and potentially revictimizing transitioning women veterans [3,9].

We know little about how and why some women veterans successfully navigate transition to civilian life, whilst others struggle [10]. We also know little about why rates of veteran suicide are higher for women veterans [11] or about the mental health needs of women veterans more broadly [12], and what role dominant masculinity military cultures might play. Also, few programs for transitioning veterans have been found effective for women veterans [3] because they have been developed largely for males, underpinned by masculinity-oriented cultural understandings of veterans’ experiences [2], which may contribute to gender inequalities [10]. Few studies have investigated the experiences of Australian women veterans.

## 2. Materials and Methods

This study used qualitative research methodologies to explore the military service and transition experiences of women veterans. The aim was to better understand the impact of military service culture on transition and post-military service needs of Australian women veterans, to inform policies and programs designed to support them. Ethics approval was obtained through the Departments of Defence and Veterans’ Affairs Human Research Ethics Committee (No. 420/22) and Flinders University Human Research Ethics Committee (No. 5577).

### 2.1. Study Population, Recruitment and Ethical Considerations

Women veterans who had separated from regular service in the Australian Defence Forces (ADF) since 2001 were initially selected as this timeframe reflects the potentially protracted nature of transition, which is acknowledged by the growing literature in this field as a process that can be protracted and over a long period of time, beyond the immediate experience of discharge from the service (leaving the front gate). Hence, the development of mental health and wellbeing issues may also be protracted, as can help-seeking for support to address those issues and their formal recognition by services and systems. It also reflects the cohort of women veterans who have served since the campaigns across the Middle East Area of Operations (MEAO) alongside men veterans, in deployed roles. However, two participants who separated from service prior to 2001 were included given their wish to contribute to the research. Participants served in the Australian Army, Royal Australian Navy (RAN), or Royal Australian Air Force (RAAF) and included all ranks and job roles. Some participants had been on operational deployments, whether warlike or peacekeeping. Inclusion criteria required that participants be over 18 years of age. We also wanted to respect the women participants’ autonomy and stance in wanting to come forward to share their experiences. Hence, a formal clinical screening tool was not used. Instead, we privileged their self-reported claims that they were not receiving acute mental health inpatient treatment or actively suicidal at time of interview.

Participants were recruited from across Australia, primarily through veteran-focused research organizations (Open Door and Military and Emergency Services Health (MESHA)), and the eight Veteran Project Reference Group (VPRG) members and their lived experience informal networks connected with women veterans. This was important, given known issues with trust in coming forward for the women veteran population.

Participants were provided with a participant information sheet about the purpose and aims of the research, what was involved in participation, a consent form, and further information regarding contact and support services if required. They were able to ask questions about the research prior to proceeding with interview, could cease the interview at any time, ask for the audio recording to be stopped, or choose not to answer particular questions. They were free to withdraw at any stage, with the understanding that once analysis was conducted, and data integrated into themes, removal of individual data would be difficult. Identifiable and personal data was removed from the transcripts and pseudonyms were used to maintain confidentiality and anonymity. Contact was carried out through the participants’ preferred means, and information about support services was provided.

### 2.2. Data Collection

Interviews were conducted by the project officer, primarily through Zoom and telephone, with time and date negotiated between participant and interviewer. One interview was conducted face-to-face. Interviews were audio-recorded, and notes taken during and/or after the interview to document impressions and contextual detail. Interviews lasted on average one hour and were transcribed verbatim by a professional transcriber. Interviews followed an open, semi-structured format (see Box 1).

Following the interview, participants were asked if they would like to view and verify the transcribed interview for accuracy, to further reflect on their comments and make additional comments; no participants chose to do this. Saturation was achieved after approximately 20 interviews.

Box 1Interview Guide.
Demographic dataLife before joining the Service and reasons for joiningExpectations and actual experience of being in the ADFViews on gender and military serviceExploration of “fitting in”Reasons for leavingExperiences and use of support for transitionExperiences for the first few months of transition and beyondExpectations and identity as a veteran and as a woman veteranImpacts of service on mental health and well-being


### 2.3. Data Analysis

Interview data were analyzed thematically by three research team members, with themes finalized by the wider research team and independent experts from the veteran community on the VPRG. Data management was supported by NVIVO 12 software [13]. A sample of interviews were independently open-coded by three project team members (project officer and two others) who then met to establish a draft coding plan. The project officer then coded further interviews, with critical insights provided by the research team who met regularly to discuss the ideas. As new themes emerged, coding of earlier transcripts was reviewed, supported by project officer fieldnotes and robust group discussion of the data. All coding and interpretation were then presented to, discussed in detail, and validated by the wider research team and VPRG before finalization of themes [14]. Analysis followed van Manen’s methodological approach [15] and involved reading fieldnotes and transcripts several times before going from parts of the text to the whole using detailed line-by-line and holistic approaches, enabling us to look in detail at description, use of language, emotion, and concepts, as well as silences and gestures. The holistic approach involved reflection on each interview as a whole.

## 3. Results

Twenty-two women veterans participated in the study, drawn from all states and territories in Australia. Participants ranged from 27 to 72 years in age (mean = 47 years; median = 50 years) and served in the military between 1974 and 2022. Over half (*n* = 13) of participants joined the military before the age of 20. Enlistment age ranged from 16 to 44 years (mean = 21 years; median = 19 years). Five enlisted as children under the age of 18, between 1985 and 1998. Thirteen participants served in the Army. Length of service ranged from 6 months to 37 years (mean = 14 years; median = 12 years). Six left the Service with rank of officer, eight separated as NCOs and seven as “Other” ranks. Ranks ranged from recruit to senior officer (actual rank would identify participant), with diverse roles that included human resources, defence welfare, operational supply, aviation, senior command, intelligence, engineering, and nursing. Most participants were partnered at the time of interview and had children with ages ranging from infant to adult; nine did not have children. The majority (*n* = 17) disclosed current mental health conditions; 12 disclosed a mental health diagnosis of post-traumatic stress disorder (PTSD). Physical injuries from service included: back, neck, shoulder, hip, and knee. Only one participant separated from service without any physical or psychological injuries (see Table 1). Detailed individual demographic detail is not displayed, to ensure anonymity.

Four key themes and various subthemes arising from participants’ experiences are listed in Box 2 and described in detail below, with direct de-identified quotes to exemplify ideas (with pseudonym, age, and length of service).

Box 2Summary of themes and sub-themes.Fitting in and managing identity within the military (Theme 1)
Reasons for joiningInitial experiences
Gender-based challenges in conforming to a masculinized culture (Theme 2)
Proving themselves “worthy”Assimilation and compromise as a survival mechanismDistancing from and ostracizing other womenCaring for others: compromising caring valuesPower and gender discrimination
Gender and consequences of disempowerment: vulnerability to abuse (Theme 3)
(Details reported in a further publication)
Separation and transition: Being invisible as a woman veteran in the civilian world (Theme 4)
Transition and the importance of preparation and supportAdjustment, disconnection, and the invisible veteran


### 3.1. Fitting in and Managing Identity within the Military (Theme 1)

A key theme was the way in which participants “fitted” into and adapted to a male-dominated and masculinized organization that functioned according to a rigid hierarchy, along with strict standards of routine, discipline, and conformity.

#### 3.1.1. Reasons for Joining

Most participants joined during their formative years as young adults and were seeking opportunities with hopes and dreams for a career as a soldier, sailor, or aviator and to serve the country. The military offered a career, sense of family or belonging, purpose, and security. Their reasons for joining were diverse, with some escaping from challenging life situations such as dysfunctional families or poverty:


*Defence was a way for me to get a stable income that was decent and also subsidize university. (Belinda, late 30s, 7 years)*



*So, I thought, I’ll join the navy cadets to meet some friends… I knew I wanted a career as such. I knew I wanted to get away from home. And so—and 17 gone. (Pauline, mid-50s, 20 years)*


Some wanted adventure, excitement, and a career different from the female norm at their time of joining:


*To see the world…to not become a statistic in the sense of small town, girl marries local, pregnant, divorced. Didn’t want to do that. (Gail, mid-50s, 30+ years)*


Others were exposed to the military from a young age, having a military family background, school cadets, or living alongside a military community, with the ambition to serve instilled long before joining:


*I was exposed to Defence from a young age, and from the age of 8 like my heart was set on joining the Air Force…everything I did was geared towards doing that…the bigger part of it for me was the pride in being able to serve my country. (Sarah, early 50s, <1 year)*



*So, I’m fifth generation military service…my dad did twenty years in the navy…both my mum’s parents were in the British Army…and then dad’s grandfather was Swedish merchant Navy and everyone before that was Army. So, I just always knew that I would join, (Katrina, early 40s, 10+ years)*


#### 3.1.2. Initial Experiences

“Fitting in” was described as a process of assimilation of a military identity beginning with initial or basic training, which focuses on building a collective team identity, physical fitness, discipline, and routine within an institutional living environment. Most participants described initially integrating into military culture because it met their needs for structure, purpose, and family and/or because they were familiar with and comfortable in a masculine environment. Both Olga and Victoria had experienced childhood abuse and neglect and had kept that part of themselves hidden from others prior to joining the military. Their initial experiences implied a sense of freedom, connection with others, and hence a strong sense of belonging:


*Loved the environment when I first got in, the camaraderie, just the freedom I experienced, I was like wow I am not having to hide away from the world. (Olga, early 50s, almost 20 years)*



*I always felt like I had to put blinkers on growing up, you know or even put myself in a box and not tell people the truth of what home life was really like. Whereas, in Defence, you know my story was just so common, they’re like yeah I joined to leave my family too… I didn’t have to hide anything… I just thrived. (Victoria, mid-30s, 5+ years)*


Other participants also loved the sense of structure and family that the military offered:


*I loved it. I love being part of that disciplined environment… And immediately you felt like you were part of the family. (Frances, early 70s, 20+ years)*


Descriptions revealed that many had adjusted to and integrated aspects of a dominant masculine culture through their experiences as school cadets or through being raised in military families:


*…because I had been through cadets it wasn’t all completely new to me, so I think if I hadn’t had that cadet background it would have been a lot more challenging than it was, but I knew how to sleep in the bush and I knew how to wear my uniform and do basic drill and stuff like that. (Heather, early 50s, 30+ years)*



*I mean I was raised by mum as an army brat and dad was ex-navy so if our rooms were untidy mum would literally rip everything out of the entire room, throw it into a pile in the middle of the room and we’d have to straighten everything out. So, the punishments that we had at recruit school compared to my mum on the warpath was really quite tame. And the yelling and screaming and stuff, it, I found it more amusing that anything else. (Katrina, early 40s, 10+ years)*



*I found it harder to get on with the women because I didn’t understand women at all. I had more to do with guys than I had with girls… I lifted weights, I had too many brothers… I would rather go out and sit around with the boys and go for a jog, or I bet I can do more sit ups than you can do, and all that sort of thing… I did fit in. (Terri, late 50s, <5 years)*


Their descriptions revealed that the short basic training met most participants’ needs with an emphasis on fitness, camaraderie, and discipline, designed to progress transition of the individual from a civilian to military identity:


*Loved it, best 11–12 weeks ever. You know got to test myself physically, mentally, I got the discipline and structure that I was craving, so there was no anxiety about what was happening the next day cos we knew exactly what was happening at what time… I thrived at recruits, I dux-ed everything. (Victoria, mid-30s, 5+ years)*


### 3.2. Gender-Based Challenges in Conforming to a Masculinized Culture (Theme 2)

Participant descriptions of their careers show that they were required to adapt and conform to the values embodied in the dominant masculine culture to have a career, acceptance, and respect of others.

#### 3.2.1. Proving Themselves “Worthy”

Many participants described employing strategies to minimize any perceptions of difference between themselves and men and to demonstrate that they could perform the same job. Throughout their interviews, there was a consistent focus on proving that they had the capacity to serve as a member of the military in the same way as a man. This focus on capacity was particularly exposed in descriptions of physical fitness and strength, proving competence with tasks, accommodating misogynistic behaviors, or adopting more masculine traits. Some strategies were consciously undertaken while others were adopted less intentionally due to acculturation.


*We were aware that it was male dominated… There were the comments… It was a bit of sink or swim, so we learnt to fit in… It wasn’t always pleasant but if you wanted to get by and wanted to have a career you had to make sure you could fit it in. (Quinn. early 50s, 20 years)*


One participant used the image of wearing a “mask” to hide her feminine identity, which she suppressed to fit into a masculinized culture. Julie, an officer, who served from the late 1990s until recently, had a highly skilled role requiring precise mental and physical acuity, in a traditionally male domain.


*I constantly wore a mask, in fact I still wear a little bit of a mask in (current workplace)… I don’t need to prove anything anymore and maybe I did or maybe I didn’t beforehand, but I always felt that I needed to behave and in fact not be a woman, you know behave and act like the men around me. (Julie, early 40s, 20+ years)*


Her description reveals how difficult but important it was for her to be perceived as integrating into the service, to achieve her career goals:


*I have always felt really lonely because I have been on the outside of the guys—all of their relationships…they all get along very easily—they are all very similar and joke and whilst I have to look like I have fitted in very easily, it’s always been me trying very very hard to fit in, but I’ve always felt like I am outside looking in wanting to be part of the crowd. (Julie, early 40s, 20+ years)*


For Julie, physical fitness was a key strategy in ensuring that there was “no gap” between her aptitude and ability and any of the men, as she focused on her capacity to perform the role by adopting normative male standards. Her focus was on demonstrating that she was the “same” as the men:


*I think military and military people value that strength and fitness, so strength and fitness I worked really hard at… I would be very observant of the way that they tackled an obstacle course because they are always faster and stronger than the women, but I observed how they did things, and I would copy that to a tee. (Julie, early 40s, 20+ years)*


The sub-theme of having to prove themselves worthy of a military identity pervaded the interviews. This emphasizes that equality and acceptance is not automatically given to women. The emphasis was on earning the respect of others by working harder in a particular job or task to prove they were as capable as men.


*You definitely had to work a lot harder. You had to sort of put in twice as much to get the same result and get some sort of respect. (Quinn. early 50s, 20 years)*


Chrissie, on the other hand, did not view her gender as an issue:


*I think it’s just about proving yourself with time. Like I didn’t, it wasn’t gender orientated…it was more about your competence and how you could look after soldiers… I saw my job as looking after my soldiers. And I think if you do that properly then your soldiers respect you no matter what your gender. (Chrissie, mid-40s, 10+ years)*


#### 3.2.2. Assimilation and Compromise as a Survival Mechanism

For many participants, fitting in meant a sensitive navigation of the military culture by learning what was both acceptable and expected behavior. This was accomplished with some compromise of their own values and/or adoption of cultural attitudes and behaviors. Again, the focus was on demonstrating that they were no different to men, to have a career. Pauline, who served for nearly 20 years and separated in the early 2000s, described her perception that women were not wanted in the military:


*But it’s kind of how I felt at the time that I had to prove to people that I’m on your side. I’m not one of the enemy. (Pauline, mid-50s, 20 years)*


Key cultural norms experienced by most participants included exposure to alcohol and misogynistic attitudes. For many, being perceived to assimilate these norms became a survival mechanism rather than part of their identity:


*Absolutely there was lots of that, there was lots of drinking, that’s what everybody was doing, so we would do that. (Quinn. early 50s, 20 years)*



*I would drink to try and keep up with the boys, to be one of the boys… At the time it was just—I don’t know it just felt like that’s how you got along with people. And I don’t think I consciously went—oh I must be one of the boys. But now that I’m older, and I look back, I actually think it was a bit of a protection thing. Like a bit of self-survival thing, you know. (Terri, late 50s, <5 years)*


Dealing with misogynistic comments or language required sensitivity and a clear strategy to manage the impact. Katrina described not reacting to comments, to reduce risk of being targeted as different, highlighting that this was a survival mechanism:


*But there was a certain level of crudity with language and jokes, misogyny. I mean if you were offended by the use of the word cunt then you’d never have survived. If guys started talking about shagging some of the other girls and you were, if you then said, “That’s inappropriate”, you would never have lived it down. (Katrina, early 40s, 10+ years)*


Some participants highlighted the need to manage or suppress their own behaviors, to conform, minimize difference and reduce the risk of being targeted. Pauline described how she downplayed her feminine side to feel accepted, compromising her sense of feminine self:


*I kind of felt like I had to try and downplay the fact that I was a girl. And I’d just try and be one of the boys… I tried to be someone I wasn’t to be accepted. (Pauline, mid-50s, 20 years)*


Katrina described being briefed and forewarned by her father who had served:


*Dad had said don’t show off, don’t talk back and all this sort of stuff so I didn’t. (Katrina, early 40s, 10+ years)*


Participants had similar recollections of compromises made by other women to conform and be accepted. Katrina’s comment highlights the difficulties for women in integrating into a culture she perceived as resistant to change.


*Watching some of the much younger girls start to adopt really crude language to try and fit in and become one of the boys, it was, it’s like watching a hermit crab put on a shell that’s the wrong size… So some of those interactions were incredibly uncomfortable for all of the women involved… I was cringing on the inside for the girl that’s trying to be one of the boys and failing or lowering herself to a standard that she shouldn’t be or for the girls that were being unrealistic in their expectations that things were going to change. (Katrina, early 40s, 10+ years)*


#### 3.2.3. Distancing from and Ostracizing Other Women

While the majority of participants had taken on the military values of strength, capability, teamwork, and discipline, their descriptions revealed some identification with the masculinized military value of stigmatizing and rejecting physical or emotional weakness in others. This was apparent in some descriptions where acceptance meant distancing the self from other women labelled as “weak”, in order not to be similarly stigmatized because of their gender. These descriptions highlight the socially constructed binary of women as “weak”: men as “strong” and how women can deny their gender differences to be perceived as the same as men in the military.


*If there any sort of weakness that’s found that’s associated with femininity you are very—there is always I think that ability for that really hard and fast wall to come down to separate from any of the assumption that you could be associated with this person, regardless of what it’s about. If it’s that you can’t do…pushups that’s because they are all women and they are weaker, and you are like oh I had better do 300 pushups. (Madison, late 20s, <2 years)*



*If there were women doing the course and they were failing at it which that was quite regular, failing… I would distance myself from those people because I felt like that is going to tarnish me as well, being another woman. (Julie, early 40s, 20+ years)*


Naomi (early 50s, served <5 years) described herself as never managing to “fit in”, lacking the skills and ability to fully conform to standards of routine and discipline. She described how she was singled out and ostracized by the other women as being weak and therefore “different”:


*And it was always me who lagged behind the other girls, making me unpopular from the get-go…the girls in my room called me ‘dumb ass’ and ‘disco’. (Naomi, early 50s, 5 years)*


Others acculturated to the military value, which considered physical or mental health issues to be a sign of weakness or failure to live up to normative standards. This was revealed in comments that highlighted difficulty for participants in accepting a mental health diagnosis or a medical discharge. Participants commented on stigmatizing themselves as “weak” and a “failure”.


*… almost like you can’t be okay with being medically discharged, like that is the worst thing that can happen to you in service… Which I held that belief too…there’s something wrong with you, hence you can’t do this [military service]. (Victoria, mid-30s, 5+ years)*


Some participants described failing to seek treatment out of fear of being seen by others as a failure. Madison described hiding severe leg injuries incurred during training out of fear that she would be stigmatized by others as a “linger” (malingerer). Likewise, Deirdre sought mental health support away from base for the same reason.

#### 3.2.4. Caring for Others: Compromising Caring Values

Participants emphasized retaining characteristics such as empathy, care, and nurturing, generally associated with being feminine traits. They highlighted these traits with their reasons for joining, such as wanting to be a nurse and/or look after military members. Commitment to both physical and emotional care for others pervaded many of the interviews. In this way, the military “family” with its focus on feeding, clothing, and sheltering members enables the feminine to be expressed through the emotional care of members done by women. Heather had a role in supporting the mental health of deployed army members, and her description reveals her care for others:


*My job in Defence has always been about supporting Defence members be it by making sure they are in the right job or be it by doing the post op psych screening or by delivering XXX training… I am part of the Army, but I am supporting people so that they are there; the most well or best person that they can be. (Heather, early 50s, 30+ years)*


Likewise, Katrina revealed care for others. She described how she would attract bullies herself but would not allow her junior soldiers to be bullied:


*I wasn’t very good at standing up for myself, I’d just take it; but, if anyone was to pick on any of my juniors, I would always defend them, and it didn’t matter who it was. (Katrina, early 40s, 10+ years)*


Caring was apparent in the comments made by the participants who had been officers. Chrissie commented on the needs of her troops and her role in providing emotional care:


*They just want to be looked after, they want to know what’s happening, they want you to communicate with them…and I think that’s where I sort of came undone in the military because I saw that as my main job, looking after people. Whereas I think some of my colleagues would, were more interested in progressing through the ranks. (Chrissie, mid-40s, 10+ years)*


Linda, a senior officer, described her perception of the difference between the leadership approaches of men and women and perceived that women leaders are instinctively more nurturing:


*I think we lead differently from men. I mean I’ve never been a yeller and a screamer. Yeah, so there is that caring element… I think we probably do lead in a more caring and concerned way… I think that we do have that more caring instinct. (Linda, late 60s, 25+ years)*


However, participants also highlighted the challenges that caring for others presents in the military institution where the masculine dominates. Belinda described how caring behavior could be interpreted as sexual, reducing her professional role and command position to one of sexual interest and objectification:


*You’re the carer…you’re the person who supports people…in my role I have, I was responsible for welfare…they tell me all of their emotional things…all of their difficulty. They tell me when they’re getting dumped or divorced, they’re telling me all of, I’m the venting person, I’m the person that supports them through everything. So, then you become this kind of sounding board for everyone and then they start to see you like well this person’s been there, this person supports me, and they take that as a bit of a well I need you… I’ve had a guy threaten to kill himself with a butcher knife because I wouldn’t date him. (Belinda, late 30s, 7 years)*


Difficulties for participants would also arise when such values of care were challenged by the institution. For some, this meant they had to compromise their own values. Linda described an activity she was required to undertake, which involved active discrimination against women:


*The other stuff I had to manage which I found particularly offensive… At the time I didn’t like it, but now I just find it so horrible. But that was the system and part of the system, so I carried out the system’s instructions. (Linda, late 60s, 25+ years)*


Belinda, also an officer, described how she felt that she never fitted into the military, refusing to compromise her values. She described a situation in which her care for others was undermined by rank. When challenged by the power inherent in the chain of command, she made the decision to leave:


*I’m authentically me and I ended up, what I did, was because my CO compromised my morals, he, they, just decision after decision that made me feel uncomfortable with my values and so I pulled the pin. Because I wasn’t going to…condone decisions that I didn’t agree with. (Belinda, late 30s, 7 years)*


Deirdre, a nurse, described how the authority and power in the chain of command clashed with her nursing values and identity. Her nursing values of advocating for patient care were challenged on several occasions. These experiences, particularly exposure to institutionally condoned medical abuse, contributed to her mental health issues including moral injury:


*There was a dude that went down with heat stroke, and I put down heat stroke because that’s what he had, and the CO was like change it, and I am like “you want me to change it to dehydration, absolutely not!” because dehydration is almost like a chargeable thing like assault. You can be charged for it because you are not hydrating yourself, you are not looking after yourself, but if you put heat stroke it’s the COs problem because he wasn’t following the…work wear…ratio, and so a lot of that stuff…even when I am medically discharged like the guilt and shame of leaving Defense because I know that I left in such a garbage … place, like I couldn’t help them anymore, and I felt bad leaving and that really contributed worse to the mental health and the transition out. (Deirdre, late 30s, 5 years)*


Likewise, Eleanor, also a nurse, had her nursing values challenged on several occasions. For Eleanor, it was a lack of professional training to perform the job, a core value in the nursing profession. Eleanor’s care for others, together with the impact this had on her mental health, is also reflected in her comments:


*I was standing up and saying ‘no, this is not acceptable, this is not safe, I will not do this’, and I was reporting incidences where people had, patients had nearly died, had the potential to die. So, I was at a point where I was in so much—my anxiety, every day I was sick, I was thinking someone’s going to die today and it’s going to be due to lack of training, it might be my fault, it could be one of the medics because they’re not getting the training, and I’m fighting for them to get the training that they need. (Eleanor, early 50s, 25+ years)*


#### 3.2.5. Power and Gender Discrimination

Overall, participants described fitting in or not fitting in to varying degrees and at various times during their service. While they had diverse experiences, finding camaraderie, acceptance, and respect was dependent on their role, postings, and the culture of the particular unit at the particular time. Unit culture was set predominantly by the senior officers, whether male or female. Most participants described difficult postings that involved men or senior females who abused power and adopted authoritarian or bullying approaches. Whilst these challenges are likely common to many male ADF members too, difficulties were commonly expressed in postings where they were targeted on the basis of gender, making it difficult for women to fit in and reinforcing their perception of feeling less valued:


*The 60% good postings, working with good people, loving the job, loving the area where I was living, and experiencing life… And the 40% were situations where just involving idiots, involving men that were just harassment, or bullying, and putting—getting caught in situations like that. And having to work with—with men who were just awful. (Gail, mid-50s, 30+ years)*



*But I mean in terms of the jobs that I found most fulfilling, usually they were the most challenging jobs and they were the ones where I wasn’t being patronized, I wasn’t being sexually harassed, I wasn’t working with misogynists. And honestly, those jobs and those highlights in my career were actually few and far between. (Katrina, early 40s, 10+ years)*


For some participants, deployments to frontline operations further highlighted their gender:


*I think before I went to Afghanistan…my intention was to go all the way. I wanted to be a general. Yeah, but Afghanistan really changed my perspective in that regard. So, I got, I transferred to the reserves when I got back and then got out, yeah…there was a lot of bullying in Afghanistan just for being a woman… There’s a lot of ego that happens when you put a whole bunch of men on operations when they’re fighting an enemy. (Chrissie, mid-40s, 10+ years)*



*We’re going to a war zone…we went over to Iraq for 6 months. Every time (name of female) and I walked into that mess to have breakfast or whatever, they’d all get up and walk out. It was just—you … made to feel like a piece of shit. (Pauline, mid-50s, 20 years)*


### 3.3. Gender and Consequences of Disempowerment: Vulnerability to Abuse (Theme 3)

Participants revealed an organizational culture that they perceived values women less than men. They described, in detail, the consequences of being less valued, having less power, and therefore being incredibly vulnerable to power imbalances because of their gender and the dominant masculine culture. Pauline described an experience of being valued less:


*So, all the other male chiefs decided that we (the women) would go back to the petty officers’ mess, which is one rank below. And—and rank and mess is very structured, and very important. And that’s why they have messes. And—and you kind of think, well okay that’s logistically okay. But what they wanted to do to make room for us was to move 3 men from the petty officers’ mess into our mess. But the 3 people they picked were our subordinates. So, to me that was a very overt way of saying, even though you outrank these blokes, we think more of—we—we value them more. (Pauline, mid-50s, 20 years)*


While Pauline’s experience occurred in the 1990s, Heather, having served for over 30 years, considered that the ADF still demonstrates that women are valued less:


*Two years ago, they actually brought in a really nice merino wool dress that doesn’t even need ironing, which is amazing, but even that, so I think a lot about gender issues…when you wear the dress your medals are optional whereas if you wear the skirt and the shirt the medals are compulsory. So for men walking around an office environment, you would have women walking around in their beautiful dresses with no medals and men walking around in their pants and shirt with all their medals on and it just creates this perception that women are in the Army but they are not really doing the stuff… So, they should have made it optional for everyone or optional for no one because it just perpetuates this idea of women as being different. Women don’t have to wear their medals, but men do. (Heather, early 50s, 30+ years)*


Several subthemes were also apparent that demonstrated perceptions of women veterans as less valued, with significant adverse consequences for women veterans. These included: experiences of misogyny, sexual harassment and assault; responses to being a mother; responses to having gender-specific health issues; and responses to having mental health issues. The impacts of military sexual abuse, in particular, are profound and have been lifelong for these particular participants. They described the emotional, psychological, and behavioral impacts of this type of trauma, which included the internalization of feelings of blame, lack of self-worth and confidence, feeling unsafe, guilt, shame, and feelings of failure. Long-term mental health issues included PTS, alcohol and substance abuse, eating disorders, suicide attempts and ideation, and sexual dysfunction. The sense of betrayal by the military institution is pervasive, profound, and impacts transition:


*The Army was war, so we were at war with the people wearing the same uniform as us…didn’t even leave Australia to get PTSD. And the person that gave me PTSD was wearing the same uniform as me. (Naomi, early 50s, 5 years)*


Impacts on life for these participants include domestic violence and difficulties forming relationships, loss of custody of children, prostitution, lack of income, and homelessness.

These subthemes are described in more detail in a further dedicated publication. 

### 3.4. Separation and Transition: Being Invisible as a Woman Veteran in the Civilian World (Theme 4)

All participants described experiencing some adjustment issues following separation from the military, whether they served for a short or a lengthy period. For most, transition involved adjustment to a world of thinking and doing for themselves as opposed to a military life of control. It also involved psychological adjustment to a civilian identity:


*I think a lot of it is not actually about the transition so much it’s just the culture behind it. You spend a lot of time being told that you are better than civilians because you are in a uniform…and it’s great to have that pride… It’s you’re out and then all of a sudden you are left in this weird space of “Why am I like them, because I am signed up, I am not a civilian, but I am not in Defense anymore”. (Madison, late 20s, <2 years)*


For many participants the nature of the separation, the preparation and the degree of control they had over decision-making also determined the ease or difficulty of adjusting to both civilian life and a different identity. Participants had described the process in which basic training had enabled them to transition from a civilian to a military identity and the various ways in which they fitted into the culture. However, the process of leaving was perceived as abrupt and final. Contemporary use of the term “separation”, rather than “discharge” or “resignation”, denotes a cessation of a relationship with the military “family”. Participants shared a variety of reasons for leaving voluntarily, including family, having achieved career goals, limited opportunities for career progression, or a combination of reasons. However, the majority also described experiencing a lack of care for them or their individual service in the abrupt cessation of this relationship:


*It’s like you are no longer providing a capability that we don’t care about you anymore… Defence expects you to give up your autonomy to fit into this culture and you know “don’t worry we have got your back, the person next to you, they would lay down their life for you, we are a brotherhood” or whatever the non-gender term of brotherhood… I think Defence makes you think that they care about you until they don’t anymore… (Heather, separated 2022 after 30+ years)*



*The biggest slap in the face was having a woman—civilian APS 2 handing me a certificate of service and thanking me for what I’ve done… I just thought to myself, this can be done so much better…right outside that door were 2 warrant officers in uniform. I said, “But if that woman had walked out that door, and said to one of those guys, can I just get one of you to come in and present this certificate to this woman, she’s leaving, she’s a WO 2. Do you know how much better I would have felt about that?” Because it was coming from somebody in uniform. (Gail, separated 2022, after 30+ years)*


Few of those who were medically or administratively separated had any control over the decision to leave the military:


*I didn’t know, I was just like a zombie. I was just like that traumatized… I didn’t want to leave… I was forced to go without even knowing it…it was like, ‘get out we don’t want you, we’re getting rid of you. We don’t like females in the army… I had no support person, no one to counsel me… It was just like, you’re gone (Naomi, early 50s, 5 years)*



*So that was an awful shock… I did appeal, but nuh, they wouldn’t have a bar of it, so that was that…one day I’m in, next day I’m out. (Frances, early 70s, 20+ years)*


#### 3.4.1. Transition and the Importance of Preparation and Support

Descriptions of formalized transition support varied depending on the year in which the participant separated. Many described either no support or very limited assistance, reinforcing the abruptness they attributed to their separation process. Where formal transition support was provided, it was described as primarily male-focused, not targeted to individual needs, and inflexible. Descriptions also revealed the focus of formal transition support for these participants was on the tangibles rather than psychological adjustment to leaving the military:


*I did a one-day seminar that basically talked about basic things like how to manage money and how to write a resume… I did need it; it just aimed at the wrong level. When I was an officer, I had all this experience, I had no idea how to communicate it and they talked to me as though I had been a truck driver and so it is a whole lot of stuff…pitched at just a completely different level to where I was. (Belinda, late 30s, 7 years)*


Along with the transition seminars, some of the narratives revealed the replication of the masculinized military culture in delivery of transition programs. This was evident in the degree of inflexibility in meeting individual needs along with a focus on rules and policies:


*I applied to join the [name of workplace] and my application went a lot faster than apparently what they usually do…and then with my military background and everything they wanted me to join really quickly…and I had to get a truck license. One of the [Career Transition Assistance Scheme] CTAS things was you can apply to CTAS to get those kind of things covered of course, and that’s $1000 and/or even more… I didn’t have enough time, to apply for CTAS in advance with their 8-week notice or whatever they needed to do and get it approved before I did my truck license… I just applied in retrospect, and they wouldn’t have a bar of it. They just said ‘absolutely not, we don’t do these things in retrospect’, and I think that is not fair. Like I couldn’t control that bottom line, and it was absolutely essential for my new employment. (Olga, early 50s, almost 20 years)*


Descriptions revealed the domination of the transition process by men and how this then impacted on recognition and responding to the needs of women. As with other aspects of military culture, women were expected to conform with a system designed for men. Rochelle separated in 2022 and had worked in the transition area, which implemented the plans for how a member transitions, plans for work and study, together with entitlements. Her description reveals a potential disconnect between age, gender, and transition experience:


*So, everybody there when I was there, were in uniform serving…they were older men, but they had transitioned out and they were reservists, and they would come back on a contract…they had experienced the transition process, but predominantly were still working in the military. So, to some degree they were just topping up their retirements with tax-free money, basically…when I served there, I was the only female and there was three, four men. (Rochelle early 30s, >5 years)*


Some participants commented on the inappropriateness of the transition system especially for women separating with a history of sexual abuse, poor self-confidence, and low self-esteem. Deirdre described how, having experienced severe bullying and harassment, she needed a gentle approach focusing on building her confidence and self-esteem but did not feel safe to emotionally connect with any of the men involved in her separation:


*The discharge process I think leans more towards the males, and that’s the thing because there are so many males running the discharge process, it doesn’t suit females per se because females are more, I don’t want to say emotional because that doesn’t describe the fact that we’re more sensitive, we’re more in tune…they really self-blame and like I dedicated 22 years and I didn’t get acknowledged and the guys are all right. It’s such a different thing. (Deirdre, separated 2020, late 30s, 5 years)*


While the majority of participants experienced limited or no formal support, several commented on the importance of being prepared for leaving with a goal and purpose in life, having a support network, and having a degree of control over the separation process itself. It is notable that these participants had the skills, knowledge, and experience from their roles to accomplish this:


*l really knew, and had my network set up anyway. And that’s because, having worked in DCO, I knew to set up my network. I knew to get onto the places that I wanted to make myself known at. And I knew that I could reach out to Soldier On, or Mate for Mates, or whatever. (Gail, mid-50s, 30+ years)*



*I was like, seeing lots of people transitioning out and seeing at all different stages, and some were quite distressed, some were quite happy, others really had no direction, so I, at that point decided that I needed to have a transition plan for myself, and I started my study again. (Rochelle early 30s, >5 years)*



*It would have taken me two years because I started looking before I knew I was going to get out, but I didn’t want to get out without the certainty of new employment. (Julie, early 40s, 20+ years)*


While Victoria stated that she had no control over the decision to medically separate her, she described using her informal support network to have as much control as possible over the actual process itself. She described how she was proactive in using her peer network to learn from the experiences of others who had been medically separated with mental health issues:


*A lot of my previous colleagues that I‘d met, you know around the country and worked with in different capacities, a lot of them had, you know been diagnosed with PTSD and were discharged for mental health…so I watched all their medical discharge journeys. And then when I sensed mine was coming, I reached back out to them and said ‘What do I do, I think this is coming whether I like it or not, what can I do now?’… I started 3 months before the decision to me, but I knew it was coming, and so they said “Right, here’s what you do’… They helped me set up the right formal people, so the right military. (Victoria, mid-30s, 5+ years)*


While Victoria pulled together her support network, Deirdre’s description illuminates the impact of a poorly managed separation on her mental health and sense of identity. Her recent transition was devoid of support, goals, and future purpose. Her description of the process illuminated the lack of care and loss of identity she felt:


*It was the transition that nearly killed me. I nearly suicided because of that f…g transition, like being left at home to my own devices, that was the worst thing they could have done to someone… from middle of November 2019 until I discharged December 2020. I sat at home and then because my mental health, like I was catatonic… I don’t know who I am as a person… I was going to be a lifer, and for the first time in my life I didn’t have a plan C, D, or E…no-one called, in that transition process when I stayed at home…there was no duty of care, no reaching out, nothing…no one should be left at home, but females definitely need to sort of be socialized a little bit more. (Deirdre, late 30s, 5 years)*


Several participants highlighted, in retrospect, the difference that having support, particularly peer support with lived experience, could have made to their transition. Most of these participants suggested that this peer support both before and after the separation process would support transition and reduce the abruptness of the process:


*I think if we had almost like a—a mentor type of role… I think I would have found it easier to talk to someone in my own age range, and my own sort of background experience. (Gail, mid-50s, 30+ years)*



*I think to get to mentor someone and then chat to them every now and then about before they leave and seeing that they’ve left to see how they’re getting on… Sort of peer support…somebody who’s been there and done that I think is invaluable. (Linda, late 60s, 25+ years)*


#### 3.4.2. Adjustment, Disconnection, and the Invisible Veteran

The level and nature of preparation during the separation process was reflected in the ways in which participants revealed adjusting to civilian life. Several pursued careers and/or studies that reflected their key military roles and interests. In this way, they transitioned to a new purpose but still connected with their service experiences. Two with backgrounds in social and psychological support services commenced social work degrees before separation and are working in similar areas. A couple moved into employment in similar male-dominated occupational areas in the emergency services; others started businesses focused on veteran support. However, all revealed at least an initial adjustment process, recognizing a misalignment between ingrained military thinking, values, and behavior with those in civilian work and society. Participants described their transition experiences in detail. This included their initial sense of disconnection and ongoing experiences of disconnection. Sarah described the long-term impacts on her self-esteem of both the military sexual trauma and the military institutional betrayal in rejecting her. She described how she then normalized bullying behavior in the civilian workplace.


*I was incredibly compliant in every job that I ever did to my detriment which injured me more. So, I would put up with shit that no-one should have to put up with, and I would stay there for way longer than anyone ever should because I had no self-worth, and you know I just thought this is how bosses just speak to employees… It’s that authoritarian style and so I was kind of a slave to that for a very long time. (Sarah, 50 years, <1 year)*


They particularly noted how they felt invisible to others. Belinda described the sense of invisibility she has experienced when wearing her service medals. Belinda further described the difficulties she experienced with potential civilian employers, highlighting a lack of understanding of the skills obtained in her service.


*Certainly, if I ever wear my medals, which I don’t do very often, I’ve been asked “are they your grandfather’s medals?” or “where did you get those from?” (Belinda, late 30s, 5+ years)*



*I have been managing a hundred and fifty people and I’d been in a warzone and I have got my university degree and I’d done all these things and then I was applying for basic HR jobs because you can’t even, you can’t, I literally had people in my interviews asking me “So you can hold a gun, what else can you do…you just can’t tell people what to do here?”. (Belinda, late 30s, 5+ years)*


Twenty-one participants disclosed service-related injuries, the majority of which were for psychological issues, predominantly PTS. Whilst a few participants described positive interactions with DVA (the organization tasked with considering veterans’ compensation claims and providing support post-transition), the majority of participants describing protracted and difficult claims processes because their healthcare needs were largely invisible to and misunderstood by the various services and support systems. Katrina described the arduous process of claiming for the impacts of sexual abuse.


*My lawyer, they were saying that DVA probably wouldn’t, but I did get, they did accept female sexual dysfunction…when I was doing it on my own it was a nightmare. I thought the ones that I did do myself were pretty straightforward, but I found out that they weren’t and so then I engaged [law firm] when I got the first responses from DVA…it took them about two years to respond…every year I would get sent to medico-legal specialists all over again. Because each year it was…”is the major depressive disorder and anxiety permanent and stable. No, come back in twelve months’ time”. And so they did that four years in a row until the legislation changed to say now you can use your own psychiatrist….as the legislation changed I was with the same psychiatrist that entire time, he filled in the paperwork and then that was it. (Katrina, early 40s, 10+ years)*


Participants described how their physical health care needs were also poorly understood by service providers in the community. Pauline had described some of the impacts of long-term service from a young age on learning life skills. She commented that her understanding of gender-specific health was limited and had accessed DVA services for this information. She described the programs offered as focused primarily on men’s health.


*It was probably a couple of years ago. I did the Heart, Health Program which DVA fund… I felt like I got something out of it. But as part of the wash up to that he said, is there anything else that you think should go in there? And I said, “yeah you’ve got nothing about women’s health, or menopause, or anything like that”. And he’s like, “oh well I could probably talk to someone about that”…But there was nothing specifically for women’s health. (Pauline, mid-50s, 20 years)*


Participants also experienced invisibility with community ex-service organizations. Anna described how her needs for counselling could not be met by Open Arms (counselling service dedicated to veterans) due to lack of provision of both child-care and breast-feeding facilities and she was advised to come back when her children were older. Anna described similar difficulties with ex-service organizations running support programs for veterans.


*Within the veteran’s space there’s organizations like RSL, Mates for Mates, Soldier On, Wounded Hero—all these sorts of groups, and I am not saying what they are doing is not good, but every single time I say “hey I see you are doing these social activities, I think they are wonderful, I think that they have a meaningful impact on a veteran’s either strengthening their body, trying to reduce pain, improve their mental health that’s all great, but why is there no creche?…So what does that do to your mental health when you are trying to do two jobs at once. (Anna, early 30s, 10 years)*


While the majority of participants described accessing conventional psychological and psychiatric therapy, they also highlighted the need for “safe” spaces that provide an opportunity for women to come together to talk, connect, and be understood as women veterans. Sarah highlighted the importance of sharing stories and how finding connections with other women veterans has helped her understand the impact of her service on her psychological health.


*For such a long time I really wanted to meet someone or a group of people to be able to share and to feel… we could bring women together and be able to tell our stories and heal. I would love nothing more really… I just feel like we need to be able to just share our stories and just understand the similarities, because for me reconnecting with the veteran community once I actually did, allowed me to understand a lot of behaviors, because all these years I’ve had these particular thought processes that were put in at the age 16. (Sarah, 50 years, <1 Year)*


For many participants, being a veteran was a source of pride, but it was also a liminal and uncertain experience, trying to find their place for their former identity in the military and their identity in civilian life.


*It’s still part of me, the immense pride I was an air force person. And I think that’s okay as long as it’s not limiting my ability to be and do other things that are important to me… I don’t feel like a veteran and I don’t, I’m not part of…nobody sees me as a veteran because I’m just a mum, right? (Belinda, late 30s, 5+ years)*


Advocacy for women veterans enables some participants to regain the sense of identity, belonging and camaraderie they had in the military.


*The people that I am transitioning, I take them to this café around the corner and it’s like it’s called the Table of Silence, and I tell them, unless you are writing a suicide note out in front of me this will never get back to your Chain of Command, your family, like you can talk to me about anything… Because I still claim them as my soldiers, they are my responsibility…this is why medical health transition is so important…the reason I identify as a veteran because of what I do now as post Defense. (Deirdre, late 30s, 5 years)*


More detailed description of these experiences is provided in Appendix A.

## 4. Discussion

Participants’ narratives revealed a strong commitment to, and pride in, their service. However, despite diversity in career length, rank and service experiences they also revealed some common challenges in their military service and in integrating back into civilian life that were highly gendered, distinct from, and additional to, those faced by men veterans. Women veterans continue to be “othered” by both a military organizational culture and a civilian society that idealizes the masculine military norm. Findings suggest that gender-based experiences are a key factor in determining whether women have rewarding service careers and a more positive transition. Key considerations arising from the themes are discussed below.

### 4.1. Masculinized Culture

Challenges for women in integrating into, and remaining accepted in, a masculinized military culture is a common theme in contemporary research internationally [3,16,17,18,19,20]. Studies have focused on the nature of the military institution as valuing sameness, as opposed to diversity or difference, conformity to a set of masculine values such as strength, stoicism, reliability, discipline and authority, and rejection of any indication of physical or emotional weakness [4,16,19,21]. Wadham et al. [4] argued that, despite reform attempts by the ADF to become more inclusive of women, the impulse for all members to conform to normative masculine values is culturally entrenched and predominates. Likewise, McCristall and Baggaley [19] contend that the culture within the Canadian military promotes hegemonic masculinity, sexism, and inequality, thereby restricting inclusiveness and acceptance of difference. Daphna-Tekoah et al. [22], in a study involving women veterans from the US and Israeli militaries, described the issues facing women as a “double battle”, having to both carry out military duties and to integrate into a hyper-masculine environment. This analogy of two battles was also a finding in a study with women veterans in the US, described as fighting a war on two fronts [16].

The additional work for women in gaining acceptance by working twice as hard, having to prove their worth as equal to men, is a pervasive finding in the literature [16,17,20,22,23]. Wadham et al. [4] (p. 271) highlight that “women’s participation has been historically constructed as a matter of capacity to do the work of male military personnel”. This focus on capacity to do the job was common to the participant narratives, rather than arguments around equality and diversity, reinforcing the dominant organizational attitude that women need to prove they have the capacity to serve in the same way as a man [4]. It highlights a dominant cultural perspective across Western militaries that women are physically and psychologically less capable than men [23].

The finding that women veterans learn what is culturally acceptable behavior in order to gain acceptance is consistent with the research literature. They have to work out for themselves what strategies to use to be accepted and express their identity in a manner that enables them to assimilate and be accepted [16,17,19,24]. For some, such as “Julie” serving in a hyper-masculine unit, identity is disguised or suppressed by wearing a “mask”, a similar finding reported in other studies [19]. “Julie” and some other participants also revealed the strategy of “mirroring” [24] male behavior, observing male behaviors then conforming to their standards by adopting masculine traits and habits.

The finding that women veterans might distance themselves from behaviors considered feminine is again commonly reported in the literature [20,23]. In this study, it was commonly expressed in terms of physical fitness and vigilance against any indication of weakness. Boros and Erolin [24] (p. 332) described this as a pumping up of masculinity and tearing down of femininity, which is again a cultural response in reinforcing masculinity. The careful navigation of acceptable behavior was revealed by participants in both advice they were given by (male) relatives prior to joining and in reflections on their own integration strategies. McCristall and Baggaley [19] commented that women who try too hard to fit in are seen as incompetent and weak or labelled as bossy and dominant, a finding consistent with the current study. A related finding was the stigmatization of women who do not conform to normative male standards, by other women. Women who do not perform the appropriate gendered identity experience ostracism and bullying behaviors, a finding also reported in other studies [19].

Participants in our study described ways in which they expressed stereotypical values and traits of nurturing and care but, in several cases, they were still challenged by the hyper-masculine culture, whether in traditional caring roles as nurses or in leadership roles. This finding contrasts with findings reported from a US study [3], which concluded that women in healthcare positions were treated differently and with more equality than those in operational units. While our study included participants from diverse occupational backgrounds, those in healthcare roles also experienced gender-based discrimination and abuse. The findings also contrast with research that reports gender-based discrimination as more pronounced in male-dominated trades or units [10] suggesting that the ADF is still culturally hyper-masculine despite the clear ethic of care that women veterans can offer.

### 4.2. Gender-Based Challenges in Service

#### 4.2.1. Sexual Abuse

Participants in this study described challenges in continuing to fit in and be accepted simply because they are women. The most common challenges involved experiences of abuse. While descriptions revealed experiences of other aspects of institutional abuse including bullying, medical and administrative abuse, it was sexual abuse in its many forms that was most pervasive and impactful on transition to post-military life. Sexual abuse of women in the military is a persistent finding in the contemporary international research [3,16,17,18,22]. We discuss this issue in a further paper where participants’ experiences of MST are explored in detail.

#### 4.2.2. Being a Mother and Reproductive Healthcare

Consistent with literature internationally, our study found that drawing attention to the female body through motherhood or gender-based health issues highlighted their difference from men and marked women as less-than-ideal military members [17]. In a qualitative study with four US women veterans, Boros and Erolin [24] described the discrimination experienced during pregnancy. Similarly, Eichler [17], in a qualitative study with 33 Canadian women veterans, reported that women were treated differently following maternity leave. Some participants in our study had revealed the discrimination and, in some cases, abuse received after having children along with difficulties in balancing family and military life. The military response to motherhood reinforces both the prevailing norm of the ideal warrior as masculine and a requirement for full commitment to service above all else [25]. Taber [25] commented that combining motherhood with the military denotes an unacceptable interference with the core mission, to which a member is expected to give constant and total dedication.

Within this military context, women who become mothers can be punished and devalued. Women adapt to this requirement for dedication to service by either not having children, delaying motherhood while serving, or trying to find a balance. Consistent with international research [20,25], our study found that some women who became mothers adapted to military needs while others left service. Taber [20] argued that this adaptation only serves to sustain the masculine military culture and that military organizations promote family-friendly policies that are not always reflected in practice. In this study, we found this expressed in the differing levels of support, or lack thereof, that some participants received from their commanding officers in balancing family and work. Some studies have reported that women’s experience of being mothers is tolerated more in the support rather than operational roles, with the support roles considered traditionally feminine [20,25]. However, we found that discrimination and abuse of women veterans as mothers was not role-specific but dependent on individual attitudes and use of power by individuals in the command structure.

Gender-based health issues, particularly those related to reproductive health, again highlight the difference from the male body and result in women being “othered” through discrimination and/or abuse. This finding is consistent with other studies, highlighting that women’s health issues are considered a weakness because they do not conform to the male norm [17]. Military healthcare remains designed around the needs of men. Participant experiences suggested that the system of military health care has not kept pace with the mainstreaming of women in the forces and their increased representation.

### 4.3. Invisibility in Transition

Invisibility as a woman veteran in transition to civilian life is a consistent finding across the research literature, with transition and post-military support still primarily geared towards the needs of men, with understandings of transition processes established by men’s experiences [18,26,27]. Despite the eras in which they served and the nature of their separation from services, the finding that participants experienced similar initial transition difficulties to men (such as initial sense of alienation or disconnection, feeling unprepared for civilian life, loss of military family/camaraderie, loss of purpose, managing internalized military thinking and behavior) is consistent with international literature [3,16,22,24,28]. Similarly, having agency in the separation process, access to resources, sense of purpose, and social support was protective in managing transition [28,29]. However, consistent with other research literature [1,3,17,27], women veterans experience additional gender-based difficulties due to invisibility of their service and needs within systems of support, including military transition services, ex-service organizations, DVA, and civilian society. This was initially reflected in the process of separation itself with examples of management by men and devaluing of service for not deploying.

#### 4.3.1. Invisibility to Civilians and the Veteran Community

The finding that some women veterans experience feeling invisible, their service devalued, downplayed, or not acknowledged by some civilians and male veterans, and a replication of the challenges and issues faced in service, is consistent with international literature [18]. Studies have reported that civilians are unsure how to relate and talk to women veterans, with a general assumption that veterans are men [16,17,24]. Similarly, Feldman and Hanlon [1], in a qualitative study with Australian women veterans, reported that participants experienced a lack of awareness by the general public about their service; and prevailing cultural resistance to the notion of women serving in the military in Australia [1]. Conversely, women veterans in Israel did not feel the same lack of recognition, isolation, and social disconnection, given a much larger proportion of society had served and transitioned [22]. Best et al. [26] argued that traditional gender expectations shape societal expectations about who is a veteran and these in-turn inform social practices. International research is also consistent in finding that this cultural resistance extends to the veteran community, including some ex-service organizations. Studies have found that men are less likely to have wearing of medals challenged, and more likely to be welcomed within traditional ex-service organizations [1,17,26]. Discrimination in employment is a common finding in other research with women veterans either being ostracized for not meeting gendered expectations of behavior [3,26] or having their military skills devalued [1,17,18].

#### 4.3.2. Invisibility in Veteran-Specific Services

Lack of visibility within veteran-specific services, as reflected in lack of response to needs of women veterans as mothers of young children and lack of gender-specific information, is a finding consistent with international and Australian literature. In a US study, Thomas et al. [30] highlighted that lack of planning by programs to address child-care needs remains a barrier for women veterans in accessing services. Similarly, in a study with 60 women veterans in Australia, Crompvoets [2] reported that lack of childcare and family-friendly spaces were a barrier to accessing services. A similar finding was reported by Daphna-Tekoah et al. [22] in relation to the experiences of US women veterans. This serves to reinforce that women veterans are not considered important if veteran-specific services are not attuned to their needs [23] but places women at risk of ignoring their own health care needs through lack of childcare [27].

The lack of response to gender-specific health needs of women veterans has received attention in the predominantly US literature given the VA is a direct service provider. Physical health issues include physiological stresses of service on the female body, the impacts of fitness regimes on postpartum recovery and age-related health such as menopause, and the importance of tailoring health services to the unique needs of women veterans are well-documented [17,24,27,31,32]. Crompvoets [2] reported that women veterans perceived veteran-specific services to be developed largely for men, and civilian healthcare providers as having limited understanding of the experiences or needs of women veterans. This was reinforced by similar findings from another Australian study [1]. However, despite findings from previous Australian research, there remains little available evidence of a focus on the health of women veterans within Australia’s DVA [33].

Participants in the current study had disclosed a high prevalence of mental health diagnoses due to service, particularly PTSD; this is consistent with other research [18,23,24,27,32]. Women veterans can experience similar barriers to help-seeking as men veterans, particularly in relation to self-stigma. This is understandable given the additional work performed by women in mitigating any signs of physical or emotional weakness in the military [24]. In addition, several participants described negative, adversarial experiences of seeking support or compensation through DVA, consistent with previous Australian and US research, which described it as a “fight” for recognition [2,22].

### 4.4. Impacts

Gender-based challenges that lead to a silencing, downplaying, or hiding of veteran identity are commonly reported in the literature [3,17,24,30]. In a study with Canadian women veterans, Eichler [17] reported that women who served in support units or who did not deploy did not feel that they deserved the label of “veteran” (see also [27]). In a US study, Thomas et al. [30] reported that women veterans were inclined to hide their service experiences rather than seek to justify eligibility for services. Hiding military experiences and/or military identity from others also avoids potential re-victimization through facing memories of trauma, particularly military sexual abuse [18].

Consistent with our study, Boros and Erolin [24] found that women veterans would pick and choose what identity to share with others, depending on context and situation, to deal with societal perceptions. Women veterans occupy a range of roles and express those that have most meaning at the time, (whether mother, student, professional, etc.) with the primary identity often being that of “mother” [17,24]. Crompvoets [2] highlighted that reluctance to embrace veteran status had implications for accessing existing veteran support services. For some participants in our study, silencing or downplaying of their veteran identity was due to feelings of shame and social disconnection resulting from military institutional abuse, with potential for their veteran status remaining unrecognized and unacknowledged and their needs unaddressed over many years. Advocacy is one way of reclaiming and embracing military identity and visibility as a veteran [3,17,24]. For some in our study, advocacy was a way of continuing to provide care for others. Advocacy helps regain a sense of meaning and pride, which in turn helps the person to cope with experiences in military service [3,17].

### 4.5. Needs

Supporting veterans in the transition from military to civilian life is a recent and contemporary focus for western democracies. As such, participants in our study separated from the military during time periods when services were evolving, albeit incrementally. However, regardless of their separation type, they highlighted gender-based needs for individualized and social support along with safe spaces for women veterans. This is consistent with other research [28] and emphasizes the importance of providing a peer-level understanding of the issues to be managed in integrating back into civilian society [3,18,24,27]. Other research has also reported women veterans experiencing greater feelings of worthlessness and disconnection when separated from the military without support [1,18]. Education about the challenges they may face as women veterans, rather than gender-neutral information, might also help mitigate initial feelings of disconnection.

The need for social connection and support from other women veterans, post-military, is commonly reported in the literature [2,3], as a valuable resource for minimizing negative mental health outcomes during transition [17], and as an outlet for the telling of their stories and validation of their experiences [2,16]. Given the isolating nature of military sexual abuse and institutional betrayal, along with long-term repression of feelings, it is not surprising that women veterans need this safe sharing of experiences. Studies have found that some veteran-specific services are not considered safe for women because the presence of men veterans can act as a trigger for PTS in cases of MST [17,24]. Using data from the Australian Veterans’ and Veterans’ Families Counselling Service (known as Open Arms), Neuhuus et al. [32] (pp. 157–158) reported that, despite an expressed desire by women veterans for groups tailored to women, logistic issues and low numbers were a barrier to establishing women-only services.

Previous research has also found that some women veterans address their gender-specific needs outside of the system of care funded through DVA, through peer-led retreats which enable connection with other women veterans and activities such as meditation. Crompvoets [2] commented that the need for services not provided through formal mechanisms raises the possibility that alternative services might fill the gap and potentially be more damaging to particularly vulnerable women veterans. In short, this emphasizes the need again for DVA to address the need for gender-specific services for women in safe places.

### 4.6. Limitations and Suggestions for Future Research

The study has a number of limitations. It involved a relatively small sample of 22 women veterans despite an original recruitment target of 40. Reluctance to participate in research might be due to several factors, including a lack of readiness to talk about experiences that are potentially traumatic, being socially isolated and not readily identifying as veterans, reluctance to differentiate between men and women veterans given the work done in fitting into a dominant masculine military environment, and lack of access to the Internet. Despite the recruitment flyer making no explicit reference that may infer that we were seeking women veterans with negative experiences, women with positive experiences may be under-represented; they may have been less interested in talking about their experiences.

Conducting interviews via video or phone was potentially both a strength and a limitation. It enabled women situated across Australia to participate and provided participation access to women who may have had difficulty leaving their home due to their circumstances or mental health status at the time. The privacy and anonymity afforded to these participants when discussing significant taboo and sensitive issues “virtually” was a strength. Alternatively, not being face-to-face may have limited the ability to observe body language during their exchange which may have impacted the participants’ trust to disclose more fully their experiences, and the researcher’s ability to be fully aware of potential distress or conduct a fuller interpretation of the data.

The sample was heterogeneous with diversity in service type, age, era served, length of service, rank, role, and operational experience. This heterogeneity, together with the small sample size, precluded any analysis of experiences by corps or rank or deployment. The only common link for this sample was their gender. In addition, the sample was fairly homogenous in terms of ethnic diversity, with the majority being white. Sexual orientation was not a focus and women veterans who identity as LGBTQIA+ might have different experiences and should be a focus for future research given international studies have reported their higher prevalence of mental and physical impacts from service along with social isolation and suicidal ideation attributed to concealing their sexual minority status [18].

Despite overall heterogeneity, the strength of this study was in the in-depth nature of the interviews. Previous Australian studies have focused on women veterans who had deployment experience [1,2], whereas this sample included participants who had never deployed but experienced similar gender-based transition challenges reinforcing that these are based in military culture. Future research could also focus on gendered experiences by the different services in the ADF, by different occupations, and the extent to which women are challenged by hyper-masculine behaviors on operational deployment. Given the increasing percentage of women in the ADF, which is currently at 20% overall [34], future research should investigate more contemporary experiences of service and transition to assess whether any shifts in military and civilian culture enables more positive experiences of transition.

## 5. Conclusions

In this study, we found women veterans to be a diverse group linked by not only commitment to service and the positive attributes of military training and culture but by the challenges inherent in a highly masculinized military environment, which are replicated in transition. For some women, gendered military experiences have a long-term impact on their mental and physical health, relationships, and identity. For some, there are gender-based barriers to services and support. By understanding masculinities as multiple, we have recognized particular forms of culture that must be addressed by the military as an institution, whilst building alliances with women and men who do not align with martial (hegemonic) masculinities. The ADF has recognized the need for cultural change to address gender issues; however, it has a contested history of “to and fro” on gender change, and its efforts have been largely conservative and occasionally progress [4]. The experiences of the women veterans who participated in this study, which they described as largely stemming from this gendered military culture, reinforce Wadham et al.’s [4] call for cultural change to be demonstrated primarily in the current leadership of the military. Overall, this study adds depth and understanding to the small but growing body of research pertaining to transition experiences for women veterans.

## Figures and Tables

**Table 1 ijerph-21-00479-t001:** Demographic characteristics of participants.

Characteristic	N	Characteristic	N	Characteristic	N
Age (in years) 20–29 30–39 40–49 50–59 60–69 70–79	1 5 3 11 1 1	Age at enlistment (in years) 16–17 18–19 20–24 25–29 30–34 35+	5 8 6 0 2 1	Service Branch Navy Army Air Force	3 13 6
Relationship status Single Partnered Divorced	4 14 4	Length of service (in years) Under 1 1–4 5–9 10–14 15–20 20+	1 4 3 5 3 6	Health Status Mental ill-health from service Physical injuries from service Comorbid mental and physical injuries from service Mental ill-health not from service No mental or physical injuries	2 4 14 1 1
Children 0 1 2 3 4 5	9 2 5 4 1 1	Rank upon discharge Officer Non-Commissioned Officer Other rank Not disclosed	6 8 7 1		

## Data Availability

The datasets presented in this article are not readily available due to privacy or ethical restrictions.

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
