# Peer review of "No Women’s Land: Australian Women Veterans’ Experiences of the Culture of Military Service and Transition"

_ijerph, 2024, doi:10.3390/ijerph21040479_

Round 1

Reviewer 1 Report

Comments and Suggestions for Authors

This is a timely, well conducted and in depth study into an evolving area of international concern. It adds invaluable information and source material to a developing narrative and will be of immense importance to the military, politicians and health care providers of many nations, in particular to the 'five eyes'. It has added value to most referenced works as it covers the in-service, transition and veteran status/experience of all participants. I could find no reason to suggest changes to the manuscript and recommend publication in its current form.

If I had to suggest future research or improvement I would have liked to have heard directly from the participants more on their views or suggestions as to how problems or adverse experiences could have been prevented or been supported through during the various phases of service life.

Author Response

Response to Reviewers: No Women’s Land

Reviewer 1

This is a timely, well conducted and in-depth study into an evolving area of international concern. It adds invaluable information and source material to a developing narrative and will be of immense importance to the military, politicians and health care providers of many nations, in particular to the 'five eyes'. It has added value to most referenced works as it covers the in-service, transition and veteran status/experience of all participants. I could find no reason to suggest changes to the manuscript and recommend publication in its current form.

Response: Thank you for your positive comments about our research.

If I had to suggest future research or improvement, I would have liked to have heard directly from the participants more on their views or suggestions as to how problems or adverse experiences could have been prevented or been supported through during the various phases of service life.

Response: We have revised the concluding remarks, emphasising that change to the traditional military culture is central to preventing and addressing the treatment of women veterans. Wadham et al. (2023) has been cited because their theoretical analysis of the culture and gender relations within the military provides compelling arguments for this stance: “…occasionally progress [4]. The experiences of the women veterans who participated in this study, which they described as largely stemming from this gendered military culture, reinforce Wadham et al’s [4] call for cultural change to be demonstrated primarily in the current leadership of the military.”

We have also brought some of the detail previously contained in the Supplementary file into the body of the manuscript – this contains some direct quotes from the participants about what they said they needed.

Please note, as per reviewer 2’s strong suggestion, we have removed substantial detail of Theme 3 from this manuscript and intend to provide it in a separate paper.

Reviewer 2 Report

Comments and Suggestions for Authors

This is an important subject for examining the role and impact of gender in the context of the dominant masculinized culture on women veterans’ experiences of military service and transition to civilian life. Below are areas of consideration for improvement

Abstract:  Revise the end of the abstract. The last two sentences describe why these gendered experiences impede women’s transition support needs.

2.1 Study Population, Recruitment, and Ethical Considerations: On Line 73, you write, “ Women veterans who had separated from regular service in the Australian Defence Forces (ADF) since 2001 were initially selected as this timeframe reflects the potentially protracted nature of transition and development of mental health and wellbeing issues”.  What does this mean? At first read, I was expecting to see a rationale that perhaps it has been shown that veterans experience challenges with integration X  months after separation. I say this because veterans who are recently transitioning versus those who have been in civilian life for over two decades would have different mental health needs and challenges. It is essential to understand the specific timeframe to provide appropriate support and resources for women veterans during their transition period

Line 80:  You mention that participants self-reported that they were not receiving acute mental health inpatient treatment or actively suicidal at the time of the interview. Did you have prior screening questions to determine eligibility during the recruiting process?

Results:  Is there a reason some results were in table form and others were not? Was there a significance for this?

 The authors should consider having a table with the themes and sub-themes, which may be helpful as a quick reference to follow the organization of the results.

Also, using a block quotations format so that it is clear what the participant's words are would make for a better organization would be ideal.

In terms of the pseudonyms. Did the participants choose pseudonyms for the results, or did the research team choose them?

The results are really rich!  I am afraid the richness of all this data is being lost in the density of the paper. The results could be enough for two manuscripts and make it easier to follow. Theme 3 could seriously be a paper on its own.

3.42- This is a key theme that connects to the paper's central question. Is there a reason why no quotes from this sub-theme were included in the main paper? What makes qualitative research powerful is reading participant words. For uniformity, it does not make sense that there are no associated quotes in the manuscript.

Discussion: The subheadings in the discussion section do not match the themes in the results section. Was it a deliberate choice to highlight a portion of the results in each theme? If so, this may be further indication that the authors tried to put too much information into this manuscript.

Limitations: Due to the sensitivity of the questions, was conducting all but one interview via video or phone a limitation? 

Comments on the Quality of English Language

Minor editing of English language required

Author Response

Reviewer 2

This is an important subject for examining the role and impact of gender in the context of the dominant masculinized culture on women veterans’ experiences of military service and transition to civilian life. Below are areas of consideration for improvement.

Abstract:  Revise the end of the abstract. The last two sentences describe why these gendered experiences impede women’s transition support needs.

Response: Thank you. We have added the following to clarify this point. “This can create significant gender-based barriers to services and support for women veterans during their service, and it can also impede their transition support needs.”

2.1 Study Population, Recruitment, and Ethical Considerations: On Line 73, you write, “Women veterans who had separated from regular service in the Australian Defence Forces (ADF) since 2001 were initially selected as this timeframe reflects the potentially protracted nature of transition and development of mental health and wellbeing issues”.  What does this mean? At first read, I was expecting to see a rationale that perhaps it has been shown that veterans experience challenges with integration X months after separation. I say this because veterans who are recently transitioning versus those who have been in civilian life for over two decades would have different mental health needs and challenges. It is essential to understand the specific timeframe to provide appropriate support and resources for women veterans during their transition period.

Response: We have revised this sentence to clarify, as follows: Women veterans…potentially protracted nature of transition which is acknowledged by the growing literature in this field as a process that can be protracted and over a long period of time, beyond the immediate experience of discharge from the service (leaving the front gate). Hence, the development of mental health and wellbeing issues may also be protracted, as can help-seeking for support to address those issues and their formal recognition by services and systems.

Line 80:  You mention that participants self-reported that they were not receiving acute mental health inpatient treatment or actively suicidal at the time of the interview. Did you have prior screening questions to determine eligibility during the recruiting process?

Response: We did not use formal screening questions because we felt we needed to respect the women participants’ autonomy and stance in wanting to come forward to share their experiences. We have revised the text as follows: “Inclusion criteria required that participants be over 18 years of age. We also wanted to respect the women participants’ autonomy and stance in wanting to come forward to share their experiences. Hence, formal clinical screening tool was not used. Instead, we privileged their self-reported that they were not receiving acute mental health inpatient treatment or actively suicidal at time of interview.”

Results:  Is there a reason some results were in table form and others were not? Was there a significance for this?

Response: We were primarily concerned about the length of the paper and the word count. Hence, we have provided more detailed description of some themes as a Supplementary file for further reading, should readers seek this, whilst ensuring the essence of those themes are captured in the body of the manuscript. The theme related to sexual violence that is also in a table was done so in the body of the manuscript to draw the reader to its significance. However, as per the reviewer’s suggestion, we have now removed this table and have developed a further dedicated paper on this theme.

 The authors should consider having a table with the themes and sub-themes, which may be helpful as a quick reference to follow the organization of the results.

Response: We thank the reviewer for this excellent suggestion. Please see Box 2.

Also, using a block quotations format so that it is clear what the participant's words are would make for a better organization would be ideal.

Response: thank you. We have made this change throughout the manuscript.

In terms of the pseudonyms. Did the participants choose pseudonyms for the results, or did the research team choose them?

Response: Pseudonyms were assigned by the interviewer following the alphabet and according to timing of each interview. (ie. 1st interview = A = Anna; 2nd interview = B = Belinda; etc)

The results are really rich!  I am afraid the richness of all this data is being lost in the density of the paper. The results could be enough for two manuscripts and make it easier to follow. Theme 3 could seriously be a paper on its own.

Response: Thank you for your encouragement. Yes, we had considered this initially when writing the paper and chose to present the complete data in one paper initially, expecting that reviewers would provide their expertise and advice about the best path to take. So, thank you!

3.42- This is a key theme that connects to the paper's central question. Is there a reason why no quotes from this sub-theme were included in the main paper? What makes qualitative research powerful is reading participant words. For uniformity, it does not make sense that there are no associated quotes in the manuscript.

Response: Again, thank you. We have now included key quotes in the body of the manuscript, drawn from detail in the Supplementary file.

Discussion: The subheadings in the discussion section do not match the themes in the results section. Was it a deliberate choice to highlight a portion of the results in each theme? If so, this may be further indication that the authors tried to put too much information into this manuscript.

Response: Rather than simply mirror the theme structure, we chose to focus on key considerations for discussion. We have revised the text as follows: “Key considerations arising from the themes are discussed below.”

Limitations: Due to the sensitivity of the questions, was conducting all but one interview via video or phone a limitation? 

Response: Thank you for raising this point. It was potentially both a strength and a limitation. Conducting interviews via video of phone enabled women situated across Australia to participate. It also provided participation access to women who may have had difficulty leaving the house due to their circumstances or mental health status at the time. The privacy and anonymity afforded to these participants when discussing significant taboo and sensitive issues ‘virtually’ was a strength. Alternatively, not being face-to-face may have limited the ability to observe body language during their exchange which may have impacted the participants’ trust to disclose their experiences and the researcher’s fuller interpretation of the data.

We have included this detail in the limitations section as follows: “Conducting interviews via video of phone was potentially both a strength and a limitation. It enabled women situated across Australia to participate and provided participation access to women who may have had difficulty leaving their home due to their circumstances or mental health status at the time. The privacy and anonymity afforded to these participants when discussing significant taboo and sensitive issues ‘virtually’ was a strength. Alternatively, not being face-to-face may have limited the ability to observe body language during their exchange which may have impacted the participants’ trust to more fully disclose their experiences, and the researcher’s ability to be fully aware of potential distress or conduct a fuller interpretation of the data.”

Round 2

Reviewer 2 Report

Comments and Suggestions for Authors

You did a good job incorporating the feedback into the revised version. I am glad that you will have another manuscript from this so that the rich data is not lost in the magnitude of the results section from the previous version.